# Effects of Capacitive-Resistive Electric Transfer on Sports Performance in Paralympic Swimmers: A Stopped Randomized Clinical Trial

**DOI:** 10.3390/ijerph192114620

**Published:** 2022-11-07

**Authors:** Luis De Sousa-De Sousa, Hugo G. Espinosa, Jose Luis Maté-Muñoz, Maria del Carmen Lozano-Estevan, Sara Cerrolaza-Tudanca, Manuel Rozalén-Bustín, Samuel Fernández-Carnero, Pablo García-Fernández

**Affiliations:** 1Department of Radiology, Rehabilitation and Physiotherapy, Faculty of Nursing, Physiotherapy and Podiatry, Complutense University of Madrid, 28040 Madrid, Spain; 2School of Engineering and Built Environment, Griffith University, Brisbane, QLD 4111, Australia; 3Department of Nutrition and Food Science, Faculty of Pharmacy, Complutense University of Madrid, 28040 Madrid, Spain; 4Faculty of Health Sciences, Alfonso X University, Villanueva de la Cañada, 28691 Madrid, Spain; 5Physiotherapy and Pain Group, Department of Physiotherapy and Nursing, Alcalá University, 28801 Alcalá de Henares, Spain

**Keywords:** diathermy, paralympic athlete, performance, time trials, swimming

## Abstract

Throughout history a variety of therapeutic tools have been studied as possible enhancers of sports activities. This study proposes the use of Capacitive-Resistive Electric Transfer (CRET) as a performance booster to paralympic athletes, specifically those belonging to the Spanish Paralympic swimming committee. The study was a randomized, single-blind, and observer-blind, crossover clinical trial. Six athletes were randomly assigned to three groups: one treated with CRET (A); a placebo group (B) and a control group (C). The CRET group attended a twenty-minute session before being subjected to pool trials at distances of 50 and 100 m at maximum performance. Measurements were in two dimensions: time in seconds and the Borg scale for perceived exertion. Comparisons between groups were made with respect to distance and the main variables. In the case of perceived exertion, no significant changes were observed in any of the distances; however, in the case of the time variable, a significant difference was observed between Group A vs. Personal Record at 100 m distance (76.3 ± 6.8 vs. 68.4 ± 3.3). The proposed protocol and level of hyperthermia applied suggest refusal of CRET use for the 100-m distance a few minutes before sports practice. Our analysis suggests the need to modify the presented protocol. ClinicalTrials.gov identifier under NCT number: NCT04336007.

## 1. Introduction

Capacitive-Resistive Electric Transfer (CRET) works with a frequency of 448 kHz [1]. CRET is used in rehabilitation for different joint range restoration [2] and pain control [3].

To these we can add other physiological effects, such as increase in temperature both on the surface of the skin and at a greater depth [4,5], a temporary increase in intramuscular and topical blood flow after treatment with the resistive electrode [6], and better execution of functional movements [3]. This therapy also favors venous return [7] and intensifies muscle flexibility [8] while raising oxyhemoglobin and, consequently, total hemoglobin [5].

Evidence shows changes in sports performance after modification of different variables. These variables include modality of training [9,10], location [11], load and duration, each resulting in changes in athletic performance [12], and potentially affecting the success of elite athletes in competitive events [13]. Some physiological variables also affect sports performance. Pain, injuries and a low concentration of hemoglobin affect performance negatively [14,15]. In contrast, a study with elite swimmers showed that high hemoglobin levels improve times over short distances and sprints [16]. Additionally, sprint range of motion can affect performance positively [17]. An increase in intramuscular blood flow improves oxygen supply [18] and hyperoxia consequently enhances exercise [19]. Finally, an increase in basal temperature with a limit at 39 °Celsius has shown to be beneficial for sports performance [20] and swimming in particular [21].

The following graphic shows an Ishikawa cause-effect diagram of how physiological effects of CRET can theoretically improve sports performance (Figure 1).

In paralympic swimming, athletes have different physical or non-physical disabilities (visual or cognitive) that affect sports performance [22]. Swimmers are classified using a functional classification system that assigns and integrates athletes with disabilities. The relationship between disability and swimming performance is inconsistent, which can disadvantage some athletes [23] in different performance classes. 

Measuring instruments with sufficient validity, reliability, and sensitivity are needed for assessments. Time trials are used widely for assessments, as well as time to fatigue [24,25]. Competitive measurement normally focuses on distance, time, and speed. Training should also focus on other variables such as perceived effort [26] according to the Borg scales [27].

Paralympic swimmers and Olympic athletes have the same performance pattern in speed competitions [28] and even the training schedules [29]. Swimming is affected by the type of disability [30]. Although paralympians perform the start, turn, and finish slower than Olympians, and have a slower clean swimming speed, they have a very similar pattern of sports performance [31].

Growth hormone case [32], has been used to investigate enhancement in sports performance [33], as has remote ischemic Preconditioning. This preconditioning is used as a surgical treatment to reduce markers of ischemic damage in the operating room [34], and increases anaerobic muscle performance [35]. We propose the use of therapeutic technology once more as a sports tool.

Regarding previous research on CRET and sports, a recent systematic review explored the use of this technology in the sports field [36], and identified only one study that assessed biomechanical changes in athletes and other physiological effects [37]. The review pointed out the lack of evidence for this technology’s potential in sports performance, and the need to address this gap [36]. No study has been carried to analyze the effects of this technology on variables for sports performance such as time trials or the Borg scale, and it has not been applied to the Paralympic population.

Because of the above, the scarce existing literature on Paralympics [38], and the possible theoretical benefits it would confer on sports performance, our hypothesis is that by applying CRET to Paralympic athletes prior to sports activity, performance would improve with respect to reduced times and perceived exertion compared to a control group and a placebo group As well as comparing differences between the times obtained between the three groups, we compared the results of time trials to the athlete’s best record time.

## 2. Materials and Methods

### 2.1. Study Design and Ethics

A cross-over trial with three conditions was carried out. The study was randomized, single-blind, and observer-blind, and the treatment group to the placebo and control groups. Participants attended consultations on three occasions for data collection and application of treatments between 09:00 and 12:00 a.m. There was a minimum of one postprandial hour and one-week minimum wash-out period to exclude carryover effects. They were asked to make sure that the food eaten on the days of the study was similar. On one occasion, CRET was applied (A) as a placebo with the machine on, but with no energy transmission (B), and on another occasion, no treatment was applied (C). The order in which the subjects passed through the groups was randomized individually, based on the table of random permutations of Moses and Oakford [39]. Randomization was therefore executed in a masked manner. Researchers collecting the data, as well as those performing statistical analyses, were blind to sequence allocation.

Ethical approval was acquired from the local research ethics committee (20/135-EC_PS_TFM), and studies performed in accordance with the principles stated in the Declaration of Helsinki, and its latest modification. All participants signed informed consent and those with visual impairment were presented with a braille transcription (done by “Ilunion Salud” supervised by the Spanish Braille Commission), or a virtual version (QR-code) if required.

### 2.2. Experimental Design 

An illustrative explanation of the experimental design is shown in Figure 2.

### 2.3. Participants

This study was carried out in the High-Performance Center of Madrid (Centro de Alto Rendimiento—CAR). Treatment and placebo application w8s carried out in the physiotherapy room, and simulations in the Olympic swimming pool, C/Martin Fierro, 5, 28040—Madrid. Purposive sampling was used. The population was adult athletes with physical disabilities who belong to the Spanish Paralympic Swimming Committee and/or are competing in federation events. They were approached directly by the physiotherapy department of the Paralympic committee.

Inclusion criteria were the following: (1) age 18–70 years; (2) possess previous Paralympic qualification in swimming; (3) signed the informed consent, and (4) had a recorded personal best in swimming (50–100 m). Exclusion criteria included pregnancy, use of a pacemaker or other type of electronic implant, unprotected skin (open wounds or recent burns), a known allergy to nickel/chromium, or present fever. Measurements were made during the pre-Olympic period between May and July 2021.

Regarding sample size estimation the GRANMO statistical calculator was used. The standard deviation of swimmers with disabilities is 0.08 m/s of the speed of swimming in competition according to the meta-analysis carried by Feitosa WG et al. [40]. A minimum difference 0.07 represents an improvement of more than 1 s in the total time in both distances, which was considered a significant improvement in a study conducted in 2017 [41], assuming an alpha error of 0.05, a power of 0.8 and a percentage of loss of 15%. A sample data size of 13 subjects per intervention group was estimated (each subject participates in the three intervention groups). Due to the lack of athletes competing at this level, and the COVID-19 pandemic, analysis was carried out with a sample size lower than estimated, including six participants, this number previously being used in a variety of studies with Paralympic athletes [22,42].

### 2.4. Intervention

2021 being an Olympic year, it was important to establish benchmarks at the time of the study since they could be different from personal best times. The control group represented an accurate base to use as a reference. Because CRET is a therapy applied physically by a professional, to isolate treatment effect of the technology, a placebo group was introduced. Three-intervention groups were studied. The CRET in group “A” had treatment applied by Indiba^®^ Activ CT 9 (Indiba S.A., Barcelona, Spain) equipment. This equipment was manufactured and calibrated to have an output frequency of 448 ± 1 kHz. There is the possibility of applying it in two modalities and powers: capacitive (maximum power of 450 VA) and resistive (with maximum power in 200 W). These modalities are expressed in percentages by the monitor of the device and the treatment was carried out in upper limbs, due to their importance in propulsion in sport [28]. Conductive cream was applied and a resistive electrode was used 65 mm in diameter, rigid, metallic (alloy containing nickel and chromium) and without any coating. The energy passed through the body directly to a neutral plate that was placed in the abdomen. The initial power was set at 75% and if the heat generated not tolerable, it was decreased to a power of 35%. The exclusive use of this electrode and power settings were based on published physiological benefits [6]. The treatment was applied for a period of 20 min and distributed equally to the extended upper limbs, with electrode rotation-translation movements in cycles of 1 s. At the end of the application, the cream was carefully removed with a cotton towel. In Group “B”, the same procedure was performed except that the appliance was used in capacitive mode so that the sounds, lights resembled the application of the treatment but the electrode was off. In the case of Group “C”, the participant went directly to the pool and started with their usual warm-up directly.

### 2.5. Outcome

Time in seconds that takes the athlete to complete simulations of fifty and one hundred meters in the pool were measured. The measurement was made with a digital chronometer (FINIS^®^ 3x-100 with a precision of 1/100 s). To measure the athletes’ ratings on perceived exertion, the “Borg RPE Scale^®^” was used. This was previously validated and recorded pre and post-simulation with its corresponding license.

### 2.6. Swimming Tests and Data Collection

Anthropometric and socio-demographic data were collected and made anonymous digitally by a computer. Tests were carried out in an Olympic swimming pool (50 m long by 25 m wide) with an average temperature of 28 ± 0.6 °C; PH of 7.1 ± 0.04 with a controlled environment temperature of 28.4 ± 0.1 °C. The athletes performed their usual warm-up routines, subsequently documenting sprint distances of 50 and 100 m in seconds, at maximum effort twice per distance with a 3-min break in between, and performing tests separately to avoid modifications in performance due to other swimmers.

### 2.7. Statistical Analysis

Statistical analysis was performed using SPSS statistical software (IBM Corp., Released 2013, IBM SPSS Statistics for Windows, Version 22.0. Armonk, NY, USA: IBM Corp.). The mean and the standard deviation of the quantitative variables were calculated. For comparative analysis a non-parametric test was conducted due to the low sample size [43] and the need to reduce error rates due to the small sample size and non- normality [44]. The Friedman test was performed to determine if there were differences between the time-stamps or perceived intra-subject effort (variation of response according to the group) and inter-subject effort (treatment. Post-hoc comparisons were made by Wilcoxon’s Sign Rank test (*p* = 0.05). Effect size calculation was performed using Kendall’s W coefficient for the Friedman comparison, the results being interpreted by Cohen’s guidelines as follows: 0.1–<0.3 (small effect), 0.3–<0.5 (moderate effect) and ≥0.5 (large effect) [45], and calculating the size effect for the post hoc comparison using the Wilcoxon Sign Rank test.

## 3. Results

The random allocation of patients to the treatment groups is shown in Figure 3. 

The study was stopped due to the COVID-19 pandemic and, because of the Tokyo 2021 Paralympic Games. Statistical analysis of the data extracted to date was carried out and with the aforementioned data.

Consistency among observers regarding repeated measurements of time and the Borg scale resulted in a Cohen’s kappa coefficient of k = 0.997, this being a “very good” level of concordance according to the literature [46].

Out of the six subjects, 50% were men and the other 50% women, with an international Paralympic classification for freestyle, backstroke, and butterfly, 66.6% classified as S10 and the remaining 33.3% as S8. Concerning breaststroke, 33.3% classified as SB10, 33.3% as SB9 and the remaining 33.3% as SB8. Finally, for individual mixed classification 66.6% were classified as SM10 with 33.3% as SM8. The rest of the demographic and clinical characteristic data are expressed as mean and standard deviations in Table 1.

The average time between the application of the treatment/placebo and the performance of the test was 69 ± 43 min. The main variables are shown individually in Table 2. Results are classified by distance and group. The data constitute the mean of two repeated measurements (since for each distance two simulations were performed to increase the reliability of the measurements). 

Table 3 shows the mean obtained by group on the Borg scale as pre and post measurements. Table 4 shows time means per group.

Since there was a low sample size; ordinal variables with a non-normal distribution were considered. The Friedman test was performed for related samples. Table 5 shows the results of the Friedman test and the ranges for each variable, i.e., the time in each of the evaluated distances and the difference obtained between the measurements of the Borg scale.

Perceived effort did not show any difference between groups in any of the distances valued. Regarding time, the 50 m distance showed no difference, but significant differences occurred in the 100 m test. Post hoc comparison of the groups with the most contrasting data was performed using Wilcoxon’s non-parametric test. Group A vs. personal best record data showed significance with a *p* value = 0.02, and a size effect for the Wilcoxon sign rank of r = −0.63, which, according to the Cohen’s guidelines is large in magnitude, and in concordance with the results expressed in the Kendall’s test. The CRET group with a higher mean compared to personal best times (76.3 ± 6.8 vs. 68.4 ± 3.3).

## 4. Discussion

As far as we know, this is the first study to assess the application of CRET in sports performance in the Paralympic population. The most closely related is that of Duñabeitia et al. who studied biomechanical and physiological changes in recreational runners, without confirming positive changes in sports performance but suggesting enhanced biomechanical effectiveness of the CRET as a possible benefit [37]. Dingley et al., with the same population and similar sample size to our study (*n* = 7), documented changes in the performance of Paralympic swimmers after the use of resisted dry exercises [47].

Our preliminary statistical analysis does not to have sufficient statistical power to establish an inference to the entire population with disabilities. Only time over a distance of 100 m had a significant difference using the applied technology. 

Multiple studies indicate that sports performance is positively influenced in the presence of a competitor for psychological, emotional, or motivational reasons [48,49]. Comparing treatment groups with personal records becomes very important, since this establishes a range of approximation related to the previous performance of the athlete [50]. The CRET group showed significant differences related to personal records, and because there are no studies available to compare these results with; we discuss CRET’s main physiological effect, i.e., hyperemia [1,6,51].

The CRET group as the only group to experience an increase in temperature, so this could explain the change in performance. Aligning with the possible detrimental effects on sports performance of hyperemia, the study by Wallace et al. points out that when there is a great increase in the core temperature fatigue increases and the athlete suffers a decrease in vigor [52]. A different study pointed out how the psychological perception of heat can negatively affect sports performance [53].

Mc Gowan et al. highlighted the importance of keeping the core temperature high until sports practice, but did not identify significant changes when comparing the application of heat combined with dry exercise vs. dry exercise only [20]. In the same way, the massage used in both the CRET and in the placebo group can explain why there were differences in means when compared to the control group. Moran et al. carried out a study with the application of pre-competition massage and found no significant differences, and even suggested a decrease in the time obtained in sprint tests [54]. Moreover, a partial capacity for acclimatization to heat has been demonstrated in the Paralympic population; but not to the same extent as athletes without disabilities [55].

On the other hand, literature that supports hyperemia as a favorable effect in sports performance is much greater, in agreement with our results. A study where the application of hyperthermia was performed passively through infrared application showed a significant difference in sports performance in favor of the treatment, after applying it in a prolonged manner and removal before measurements [56]. Zapara et al. used the same passive hyperthermia application protocol and confirmed the results [57]. In favor of elevating the core temperature, a study by West et al. showed better sports performance 20 min after warm-up vs. 45 min, due to maintenance of an elevated core temperature, in the 200 m test in freestyle [58]. Similarly, from a more physiological point of view, a study by Gray S. et al. determined that the skeletal musculature increases turnover of ATP and the speed of contraction of the muscle fiber when the muscle temperature is increased passively, resulting in values closer to maximum sports performance [59].

Given this data discordance regarding hyperthermia application on sports performance, a temperature threshold could exist, in which approaching the extremes improves performance to point at it negative effects occur, with respect to both extreme cold and heat [60]. This implies the possibility that our athletes could have been outside this threshold due to the short time between the application of CRET and the simulation performance of 69 ± 43 min, considering that physiological changes have been observed in the application of this technology up to 164 min after its application [51].

Congestion generated the heat has sometimes been counteracted by pre-cooling techniques as in the case of Faulkner et al., in which the clothing of cyclists was immersed in water at a temperature of 14.2 ± 1.2 °C or frozen before performance, better results being obtained by the group in which the clothing at a lower temperature was applied. This suggest that skin temperature played a prominent role on the level of the muscular or core temperature [61]. Comparing this data with the water temperature applied in our study (28 ± 0.6 °C), it is possible that this temperature was not enough to cool the heat induced at the muscular level, the skin and the core. Confirming this suggestion, the study of Soultanakis et al. pointed out that water temperatures of 26.7–30 °C did not affect the speed or biomechanics of the stroke in distances of 50 m in freestyle sprint [62].

Corte et al. highlighted the importance of assessing the use of thermography since it allows control of body temperature and therefore improves sports performance, as well as avoiding injuries [63]. In this way we suggest an adjustment of the protocol with a greater control of temperature, either in terms of applied intensity or temporality until the performance. This may have positive effects on performance; however, these assumptions require additional verification.

The results of this study are fundamental in a practical and theoretical sense. At the practical level, the CRET hyperemia protocol proposed above did not bring the sprint distance times (100 m) closer to the personal record. The use of these treatments is not recommended immediately pre-competition. Results are of great importance at a theoretical level since the information obtained is preliminary, but supports a change in the protocol, and directions for future research.

### 4.1. Limitations

The scarce literature makes it difficult to compare studies and standardize methodology. Due to the subjects presenting different characteristics it is very difficult to establish conclusions extrapolatable to the entire Paralympic population. Not knowing the physiological changes in real-time during the treatment or during the time test, does not allow establishing cause and effect. 

Due to the current COVID-19 pandemic it was not possible to reach the calculated sample size. In addition, the trial took place in an Olympic year, making it difficult to recruit the full sample size.

As a consequence of the small sample size, the resulting values did not have significant differences, and lacked statistical power to infer properties to the rest of the studied population. Furthermore, it was more likely to incorrectly accept a null hypothesis, failing to identify a possible difference between the study groups (Type II error). 

### 4.2. Practical Applications

-Basing the practical use of CRET on known physiological effects is insufficient. Not knowing its effects on sports performance and their extent might lead to questions about its use, and possible undesirable effects. This paper presents clinicians with information related to the latest proven benefits of CRET, as well as deepening the knowledge into the undesired effect of its application immediately prior to a performance at specific parameters.-A very detailed protocol was established in this study resulting from the scientific available information. Use of this protocol as a baseline to carry out modifications could be of the utmost interest to the paralympic population, as well as the methodology presented as a tool to prove those future modifications.-This study opens a broader field of study from clinic use of this technology to the sport/competitive field. The study suggests future lines of investigation, such as real time monitoring of CRET’s physiological effects in sports. There is a current need to deepen the knowledge of the following CRET’s influence on sports performance related to duration of appliance, electrical potential, location of application, temperature threshold, training or competitive frame.

## 5. Conclusions

Although the potential effects of CRET should lead to an improvement in sports performance, this study suggests that the effect may also be detrimental effect when applied to specific parameters.

Our findings do not suggest an improvement in the variables examined here in response to the application of CRET immediately before sports performance. We have yet to elucidate the possible effects on Paralympic athletes within the wide range of applications of CRET. However, the observed modifications in variables such as timing (significant), or the perceived exertion (not significant) in the treatment group, compared to personal record in the 100 m distance, raise numerous questions, since they suggest this procedure should not be used just before sports practice. Our analysis suggests the need to modify the protocol presented. While not having sufficient statistical strength due to the sample size, we demonstrate the need for relevant modifications in terms of duration of application, the mean time between treatment and performance, and the power applied. Continued research in this field with a larger population is required.

This trial was paused due to the COVID-19 situation and the stress placed on athletes by the Paralympic Games.

## Figures and Tables

**Figure 1 ijerph-19-14620-f001:**
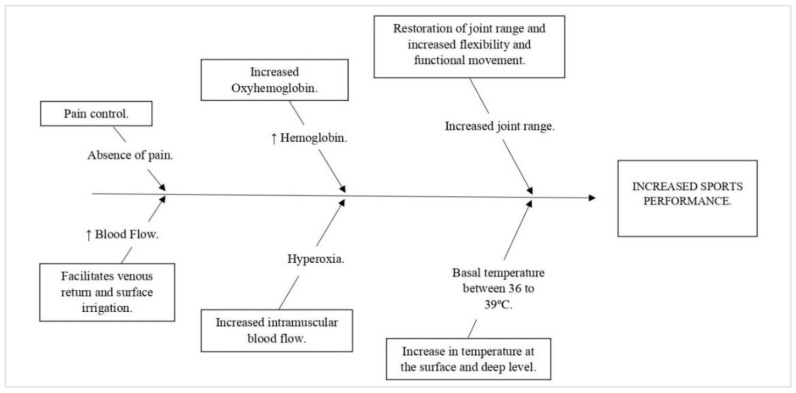
Cause-effect diagram (Ishikawa’s Thorn) of the theoretical effect of the application of CRET on sports performance, and the relationship between the effects of radiofrequency and factors that modify sports performance according to the literature. Source: Own elaboration.

**Figure 2 ijerph-19-14620-f002:**
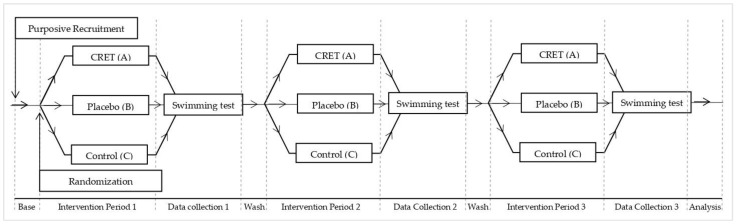
Experimental design. Source: Own elaboration.

**Figure 3 ijerph-19-14620-f003:**
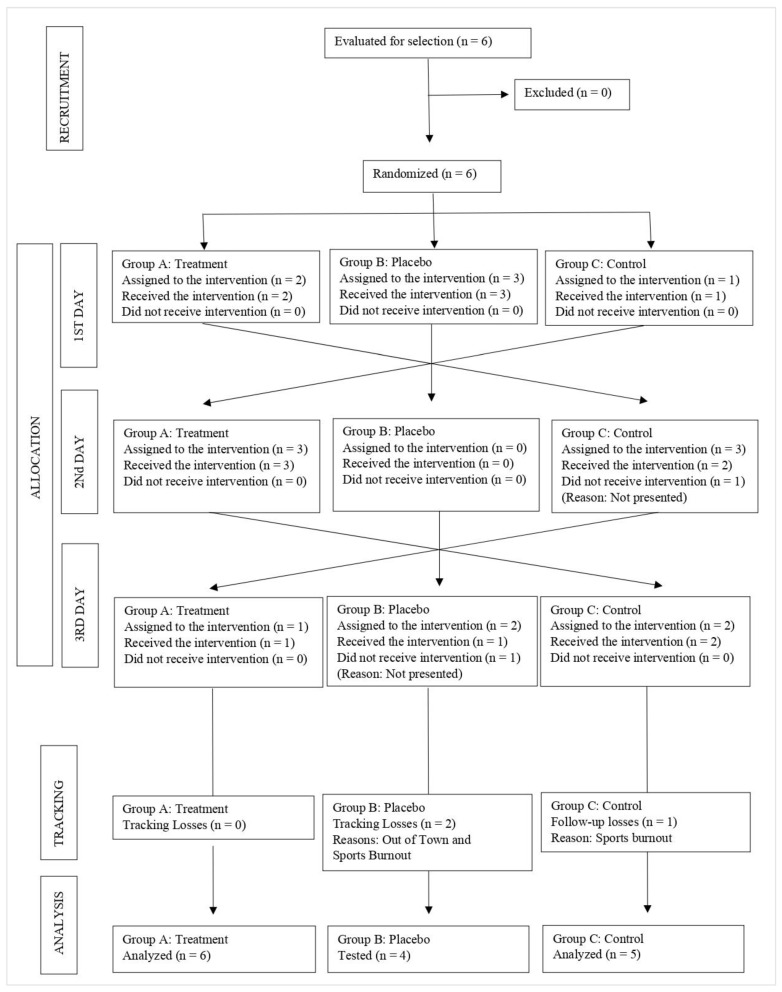
CONSORT flowchart. For each group, the number of participants who were randomized; received the intended treatment, and analyzed for the primary outcome. Source: Own elaboration.

**Table 1 ijerph-19-14620-t001:** Descriptive measures of the secondary variables collected.

Variable	Descriptive Measures
Age (years)	22.7 ± 2.3
Weight (Kg)	58.1 ± 8.9
Height (cm))	168.5 ± 11.8
Daily training (hours)	3.9 ± 0.8
Better style	
Back	28.6%
Breaststroke	14.3%
Butterfly	14.3%
Crawl	28.6%

**Table 2 ijerph-19-14620-t002:** Results per individual of their performance both in time and perceived effort in the 50 m and 100 m swimming events.

Subject	Group	Test 50 m (Average Two Tests)			Test 100 m (Average Two Tests)		
		Borg Pre	Seconds	Borg Post	Borg Pre	Seconds	Borg Post
91491a0b	A	13	34.12	14.5	12.5	72.95	17.5
	B	12	34.38	18.5	10	72.32	18.5
	C	16.5	34.45	18	15.5	74.99	19
62856cb2	A	13	39.39	15	13	86.53	15
	B	10.5	36.3	16	9.5	80.85	16
	C	11.5	37.93	13	11.5	80.06	14
b535ee0c	A	14	39.22	16	15	83.1	18.5
	B	15	37.26	18	15	82.67	19
	C	12	39.57	15	13.5	83.07	17
e867d12c	A	14.5	31.12	16.5	14	69.82	19.5
	B	11.5	31.44	14.5	11.5	68.41	16
	C	11	30.06	16	10.5	66.22	15.5
c271b626	A	11	32.58	16.5	10.5	70.81	16
	B	-	-	-	-	-	-
	C	13	33.97	16.5	14	73.1	17
1b936d0d	A	12	34.51	16.5	10	74.97	16.5
	B	-	-	-	-	-	-
	C	-	-	-	-	-	-

**Table 3 ijerph-19-14620-t003:** Mean ± standard deviation of perceived effort per group and mean difference between pre and post-measures and their respective confidence interval.

Treatment Group	Borg Scale 50 mts Test			Borg Scale 100 mts Test		
	Mean ± SD		Mean Difference (CI. 95%)	Mean ± SD		Mean Difference (CI. 95%)
	Pre	Post		Pre	Post	
A (treatment)	12.9 ± 1.2	15.8 ± 0.8	2.91 (1.17, 4.65)	12.5 ± 1.9	17.1 ± 1.6	4.66 (2.95, 6.38)
B (Placebo)	12.2 ± 1.9	16.7 ± 1.8	4.50 (1.66, 7.33)	11.5 ± 2.4	17.3 ± 1.6	5.87 (2.60, 9.14)
C (Control)	12.8 ± 2.1	15.7 ± 1.8	2.90 (1.06, 4.73)	13 ± 2	16.5 ± 1.8	3.50 (2.33, 4.66)

SD: Standard deviation, CI: Confidence Interval.

**Table 4 ijerph-19-14620-t004:** Mean ± standard deviation of the time for different distances and respective confidence intervals.

Treatment Group	Time Obtained in 50 m Test		Time Obtained in 100 m Test	
	Mean ± SD	CI. 95%	Mean ± SD	CI. 95%
A (treatment)	35.1 ± 3.4	31.55, 38.75	76.3 ± 6.8	69.15, 83.57
B (Placebo)	34.8 ± 2.5	30.75, 38.92	76 ± 6.8	65.21, 86.89
C (Control)	35.1 ± 3.7	30.59, 39.79	75.4 ± 6.5	67.38, 83.58
Personal Record	31.2 ± 1.8	29.31, 33.20	68.4 ± 3.3	64.97, 71.92

SD: Standard deviation, CI: Confidence Interval.

**Table 5 ijerph-19-14620-t005:** Results of the Friedman test. Range and *p*-value are shown to a significance of 95% by variable and distance. Test effect size calculated with Kendall’s W value.

Test Compared to Friedman	Ranks by Group				*p*-Value (Sig. 95%)	Test Effect Size
	Group A	Group B	Group C	Personal Record		
Borg Scale 50 m	1.38	2.63	2	-	0.16	0.44
Borg Scale 100 m	1.88	2.5	1.65	-	0.42	0.21
Times 50 m	2.75	2.5	3	1.75	0.55	0.17
Times 100 m	3.75	2.5	2.5	1.25	0.05 *	0.62

* *p*-Value close to <0.05.

## Data Availability

Not applicable.

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
