# Peer review of "Effects of Capacitive-Resistive Electric Transfer on Sports Performance in Paralympic Swimmers: A Stopped Randomized Clinical Trial"

_ijerph, 2022, doi:10.3390/ijerph192114620_

Round 1

Reviewer 1 Report

EFFECTS OF CAPACITIVE-RESISTIVE ELECTRIC TRANSFER ON SPORTS PERFORMANCE IN PARALYMPIC SWIMMERS: A STOPPED RANDOMIZED CLINICAL TRIAL

General Commentary

This article presents a very interesting and pertinent question of research of the investigated the use of Capacitive-Resistive Electric Transfer (CRET) as a performance booster to the Paralympic athlete, specifically for those who belong to the Spanish Paralympic swimming committee.

However, some questions need to be clarified in order to better understand and apply the results found.

MAJOR CONSIDERATION

METHODS

Experimental design

I suggest the authors add Experimental Design sub-chapter, an illustrative figure.

Participants

I did not find in the subchapter participants how and how many subjects were selected for the present study, was a sample calculation performed? if not, please justify the sample size based on similar studies in the literature.

Statistical

The description of the analysis and statistics of the data is very shallow. What was the alpha (p) value adopted a priori? Furthermore, was the data normality and homogeneity performed? In addition, I suggest that the authors present the 95% confidence interval (CI95%) in all comparisons, and finally calculate and classify the effect size of these comparisons.

RESULTS

Describe in the results of CI95% and Effect Size (ES) with their respective classification.

DISCUSSION

Limitations

Describe in the limitations the low sample N used and its implications for the results of the study.

Practical or Clinical Applications

At the end of the discussion, I suggest the authors add subchapter Practical or Clinical Applications of the results of this study.

CONCLUSION

I suggest that the authors approach the results of the present study differently in the conclusion, what was actually found? please describe directly. The other information described in the conclusion can be presented at the end of the discussion and applications.

Reviewer 2 Report

Ref. ijerph-2003720

Title: Effects of Capacitive-Resistive Electric Transfer on Sports Performance in Paralympic Swimmers: A Stopped Randomized Clinical Trial

Journal: IJERPH

Reviewer Comments to Author

Comments

Thank you for the opportunity to review the paper titled: Effects of Capacitive-Resistive Electric Transfer on Sports Performance in Paralympic Swimmers: A Stopped Randomized Clinical Trial. This study has focused on the use of Capacitive-Resistive Electric Transfer (CRET) as a performance booster to the Paralympic athlete.

The paper presents a current and relevant theme. The theoretical framework is consistent, well used, and provides an appropriate basis for the construction of the methodology. The results obtained are relevant and present a contribution to their field of study.

Introduction

The paper demonstrates an adequate understanding of the relevant literature in the field and cites an appropriate range of literature sources.

The paper demonstrates an adequate understanding of the relevant literature in the field and cites an appropriate range of literature sources.

Please specify the gaps in previous research and cite corroborating documents

What is the uniqueness of this study? Overall, the manuscript is lack contribution to the literature.The references were used to support the research objectives are supported by the latest research papers.

Materials and Methods

Please provide additional information on the reliability of the measurement data

a. What type of sampling technique was used in this study?

b. How were the participants approached?

c. How the authors control the participants?

d. When was the data collected?

e. How was the basis for the study participant grouping?

Results

The results were clearly presented and properly analyzed. The conclusions adequately brought together all the other elements of the study.

Can the experimental design procedure further supplement the basis of the group?

Discussion

There are several broad arguments and claims in the manuscript.

Particularly I encourage the authors to highlight the research’s theoretical and practical implications for each hypothesis than simply stating previous research’s suggestions.

The discussion is sufficient, but the conclusions scant and vague. This section should be expanded and deepened.

The limitations of the study and future lines of research should also be exposed.
